# First discovery of parasite eggs in a vertebrate coprolite of the Late Triassic in Thailand

Thanit Nonsrirach[1]*, Serge Morand[2,3], Alexis Ribas[4,5], Sita Manitkoon[1], Komsorn Lauprasert[6], Julien Claude[7,8]

**1** Palaeontological Research and Education Centre, Mahasarakham University, Kantarawichai, Mahasarakham, Thailand, **2** MIVEGEC, CNRS – IRD – Montpellier Université, Montpellier, France, **3** Faculty of Veterinary Technology, Kasetsart University, Bangkok, Thailand, **4** Parasitology Section, Department of Biology, Healthcare and Environment, Faculty of Pharmacy and Food Science, University of Barcelona, Barcelona, Spain, **5** Institut de Recerca de la Biodiversitat (IRBio), Universitat de Barcelona, Barcelona, Spain, **6** Department of Biology, Faculty of Science, Mahasarakham University, Kantarawichai, Mahasarakham, Thailand, **7** Institut des Sciences de l'Évolution de Montpellier (ISEM), Montpellier Université, UMR UM/CNRS/IRD/EPHE, Montpellier, France, **8** Department of Biology, Faculty of Science, Chulalongkorn University, Bangkok, Thailand

* thanit.nonsrirach@gmail.com

## Abstract

A paleoparasitological investigation of a vertebrate coprolite from the Huai Hin Lat Formation (Upper Triassic) was carried out. Five morphotypes of potential parasite eggs or sporocysts were identified in the coprolite by microscopic analysis using thin section technique. The rounded or oval shape and thick shell of one of the five morphotypes suggests that it belongs to nematode of the order Ascaridida. Systematic assignment of other morphotypes cannot be done in detail but suggests that the host was parasitized by different species of parasites. This is the first record of parasites in terrestrial vertebrate hosts from the Late Triassic in Asia and it provides new information on parasite-host interactions during the Mesozoic era.

## Introduction

Paleoparasitology is the investigation of parasites found in paleontological and archaeological sites [1–9]. Although specialized parasites producing traces in hard tissues have sometimes been partly identified [10–16], other parasites have a very poor fossil record because the soft tissues of the host in which they occur are rarely preserved, except in exceptional conditions such as in amber [10, 11]. Another important source of parasite remains are coprolites, i.e., fossilized faecal material, which can shed light on trophic chains [4, 8, 17–19].

Several reports have described parasite eggs in coprolites [3–6, 8, 18]. The assignment of coprolites to a specific host is challenging, but their different shapes can provide systematic information [20, 21]. The insights into parasite-host interactions derived from these discoveries allow a better understanding of palaeo-coevolution and palaeo-ecosystems [6, 17–19].

Helminth eggs have been described in vertebrate coprolites, including those of Permian sharks [6], cynodonts, dinosaurs [5, 8] and other Mesozoic archosaurs. and Quaternary mammals, e.g., hyena, deer and sloth [22–24]. We report the discovery of parasites, including helminth egg, in a Late Triassic vertebrate coprolite from Northeast Thailand. The coprolite was

**Funding:** This research project was financially supported by Mahasarakham University, by the

IRN paleobiodivASE CNRS, by the Hubert Curien PHC project trophic network evolution of the Mesozoic of Thailand and its impact on biodiversity (2016-2017) and by the French Agence nationale de la recherche (project ANR-11-CEPL-0002 BiodivHealthSEA). The funders had no role in study design, data collection and analysis, decision to publish, or preparation of the manuscript.

**Competing interests:** The authors have declared that no competing interests exist.

collected by a Thai-French joint paleontological field survey in 2010 during a field work in the Huai Nam Aun outcrop near Nong Yakong village (Khon San District, Chaiyaphum Province, Thailand). The coprolite is preserved in the collections of the Palaeontological Research and Education Centre, Mahasarakham University, Mahasarakham province in Thailand under the catalogue number PRC 021.

The Huai Nam Aun outcrop contains various beds of limestone and mudstone, deposited in brackish water or freshwater and in a low energy depositional environment [25]. The vertebrate fossil remains in the outcrop consist of *Hybodus* teeth, ganoid fish scales, and temnospondyl fragments [25–27]. The Huai Nam Aun outcrop is part of the Huai Hin Lat Formation (Fig 1), which has been dated as Carnian-Norian based on palynomorphs, plant macroremains, conchostracans, and vertebrate remains [28–32]. So far, the vertebrate fossils found in the Huai Hin Lat Formation comprise *Hybodus* sp. [25, 33], ginglymodians [25, 34, 35], lungfishes (possibly *Ferganoceratodus* sp.) [36, 37], temnospondyls (including *Cyclotosaurus* and Plagiosauridae) [25, 27, 38, 39], the primitive stem-turtle *Proganochelys ruchae* [40, 41],

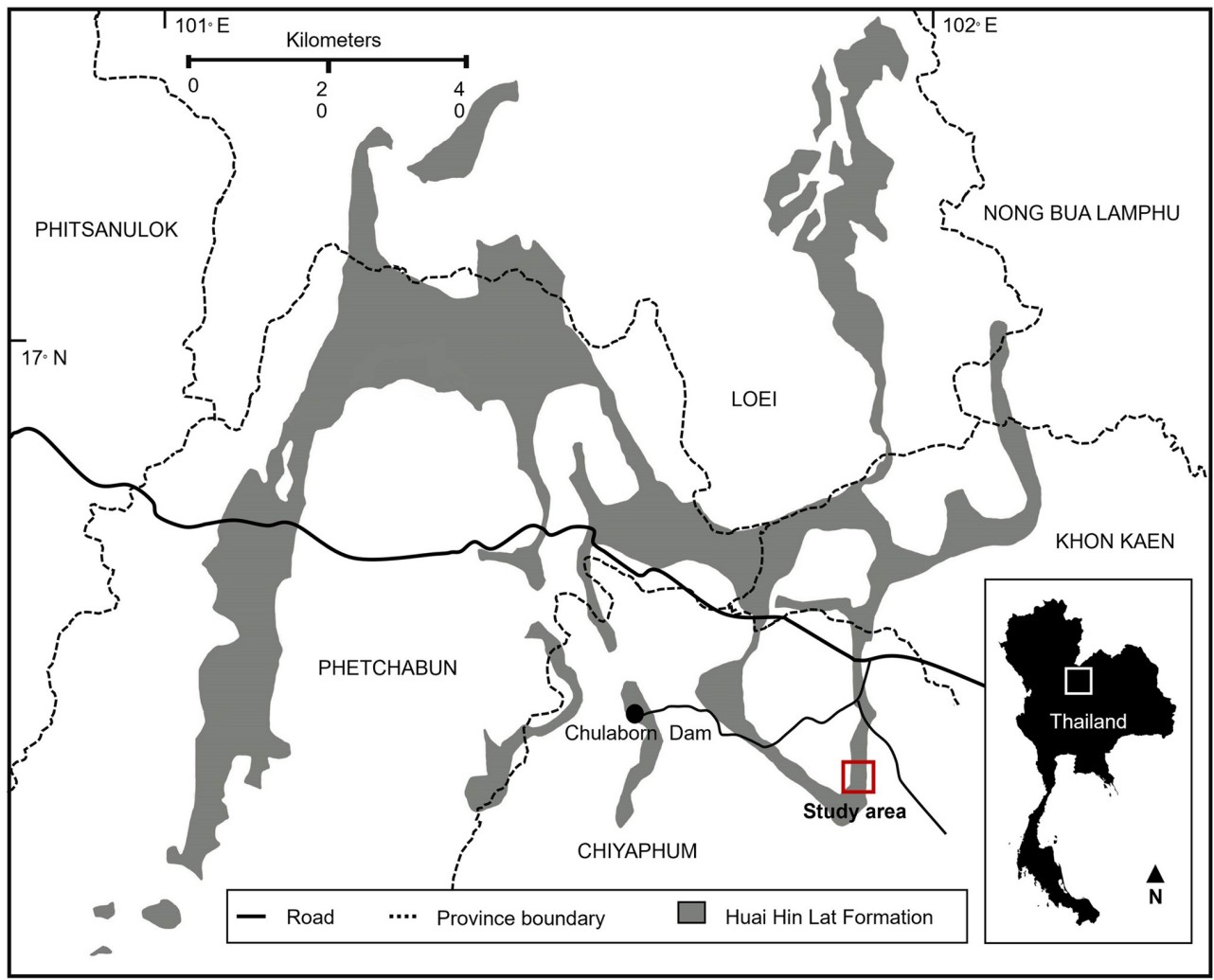

**Fig 1. The Huai Nam Aun outcrop in the Huai Hin Lat formation of Thailand [28].**

and phytosaurs [42]. Furthermore, archosauromorph footprints were identified as cf. ichnogenus *Apatopus* sp., and could have been done by a phytosaur [43].

## Materials and methods

The studied coprolite was photographed, measured, and classified based on its shape. To search for internal structures and fossil inclusion, the coprolite was hardened by embedding in epoxy resin, and then cut with a diamond saw in longitudinal and transversal sections using a standard thin section method. The coprolite slices were glued to glass slides, and optimal thickness for transmission microscopy was obtained using a grinder with a graded series [6, 44]. All microscopic structures and fossil remains were photographed with a light microscope Nikon ECLIPSE E200, and multiple images taken with different focal distances were combined using a focus stacking technique.

## Results

The coprolite has an elongated cylindrical shape, curved on one side, with a rounded end (Fig 2), and is approximately 74 mm in length and 21 mm in diameter. The surface is hard, smooth and grey in colour. Microscopic observations of all slides showed a dark, high-density clay-like material and absence of soft tissue, e.g., folded or spiral traces. Five different morphotypes of organic structures were visible in the coprolite slices.

Morphotype I: ellipsoid in shape with a round to oval sections (Fig 3) and 40 to 60 μm wide and 50 to 70 μm long. They exhibit a hardly discernible and relatively thin shell (1 to 2μm) and

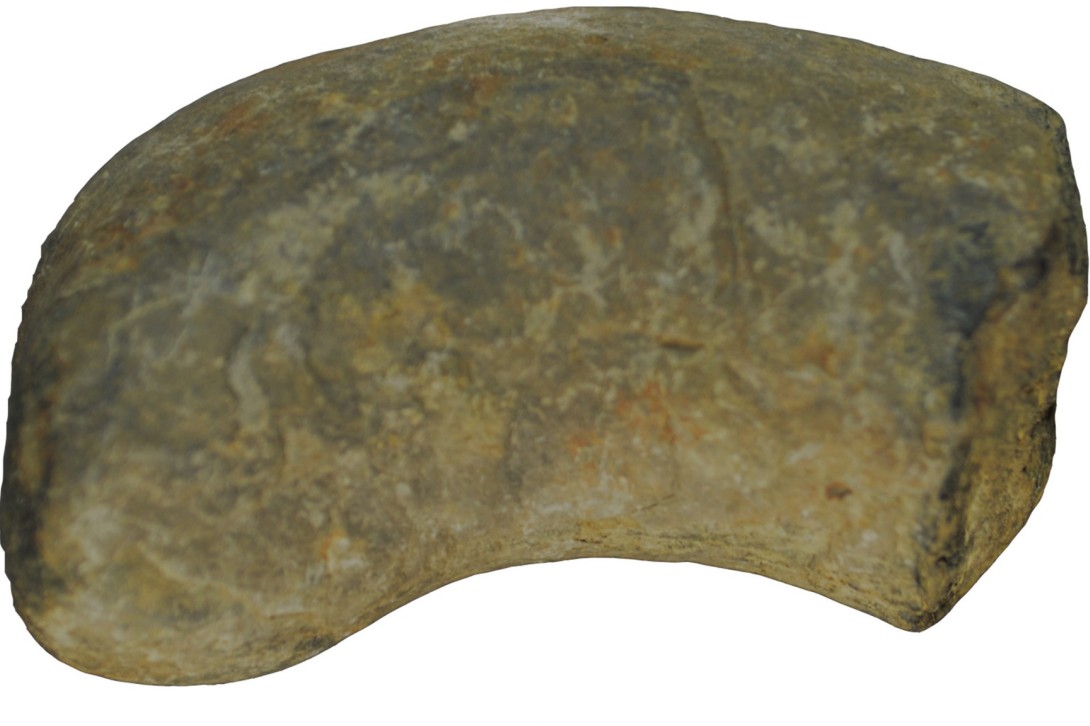

2 cm

**Fig 2. The vertebrate coprolite with parasites found in the Huai Nam Aun locality (Upper Triassic).**

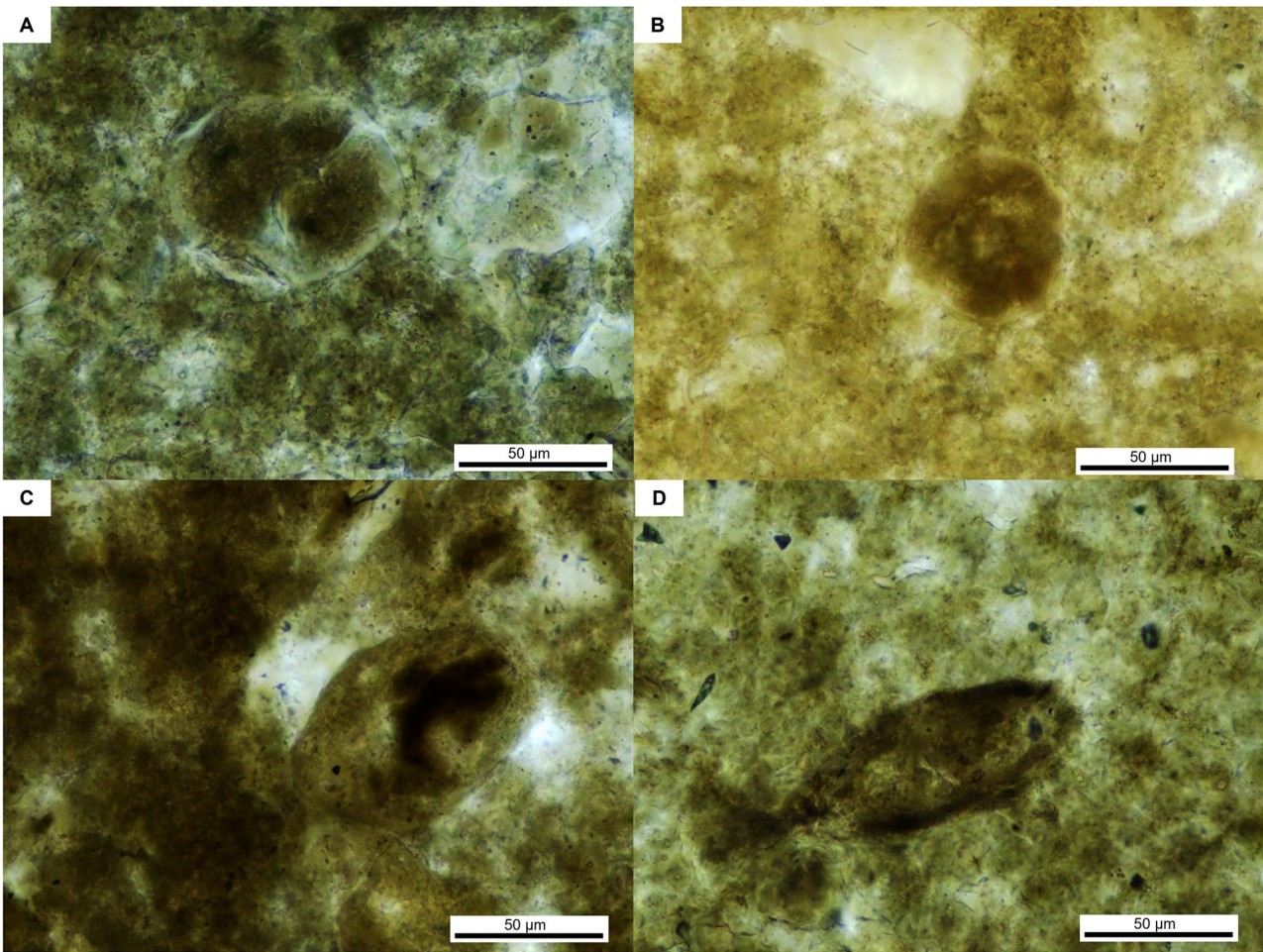

**Fig 3. Parasites of morphotype I, found in the vertebrate coprolite.**

show internal structures that could correspond to dividing cytoplasmic or nuclear material (Fig 3A). The most elongated one (Fig 3D) shows an apical opening (micropyle or operculum).

Morphotype II: spherical in shape (Fig 4) with a diameter of around 80 μm and a thick (3–4 μm) and irregular shell that is interrupted and could present a pore; the surface shows slightly developed wrinkles.

Morphotype III: irregular ellipsoid shape (Fig 5) with a minimum diameter of 80 μm and a maximal diameter of 120 μm) with a very thick shell (10 μm) and it shows a segmented or multicellular body within the shell.

Morphotype IV: spherical rounded shape (Fig 6) with a clearly defined shell of 3 to 5 μm, with a reticulated and anastomosed surface ornamentation. This morphotype is 80 to 140 μm in diameter. Within that structure, no clear cellular or nuclear material can be observed and there is no evidence of pore or operculum.

Morphotype V ellipsoid shape (Fig 7) with a very thick shell (7 to 9 μm). It is 60 μm long and 43 μm wide. In this morphotype, a second translucent layer is observable within the shell and is irregular in thickness (2 to 7 μm). There is no evidence of operculum, and the ornamentation of external surface is not properly appreciated.

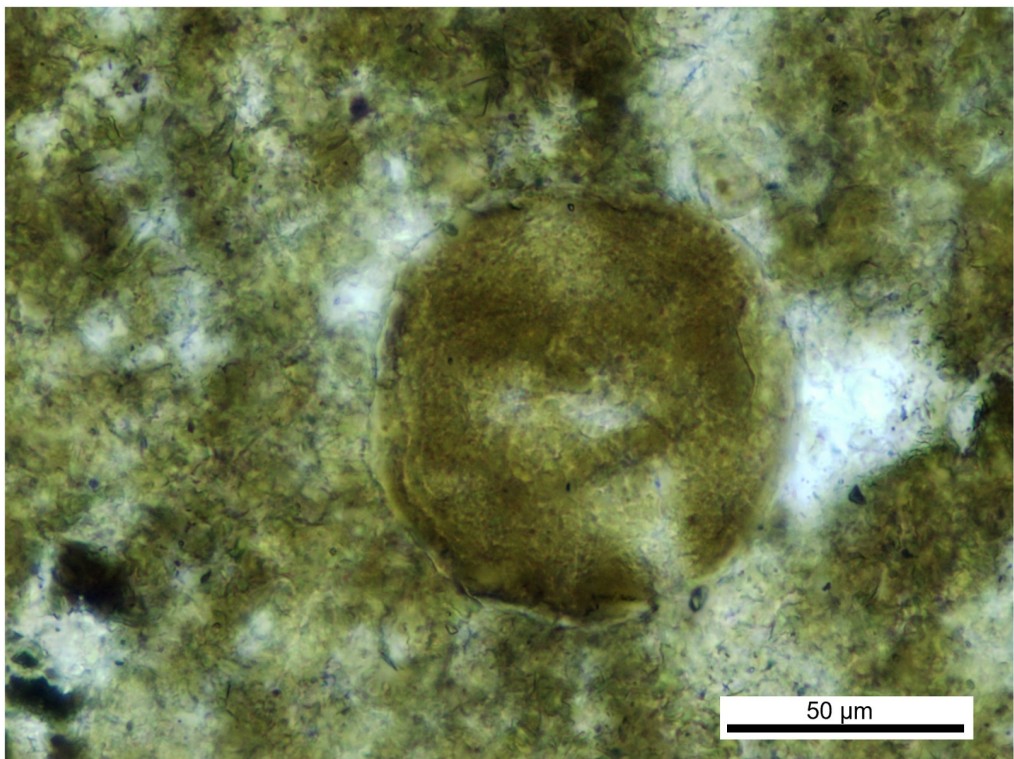

**Fig 4. Parasite of morphotype II, found in the vertebrate coprolite.**

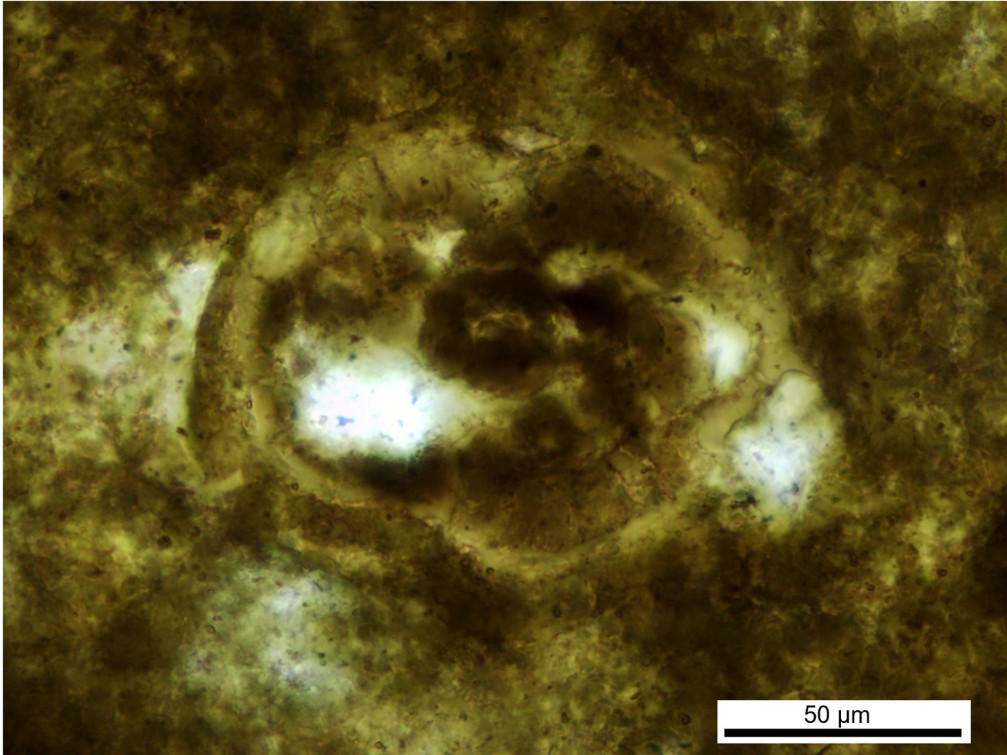

**Fig 5. Parasite of morphotype III, found in the vertebrate coprolite.**

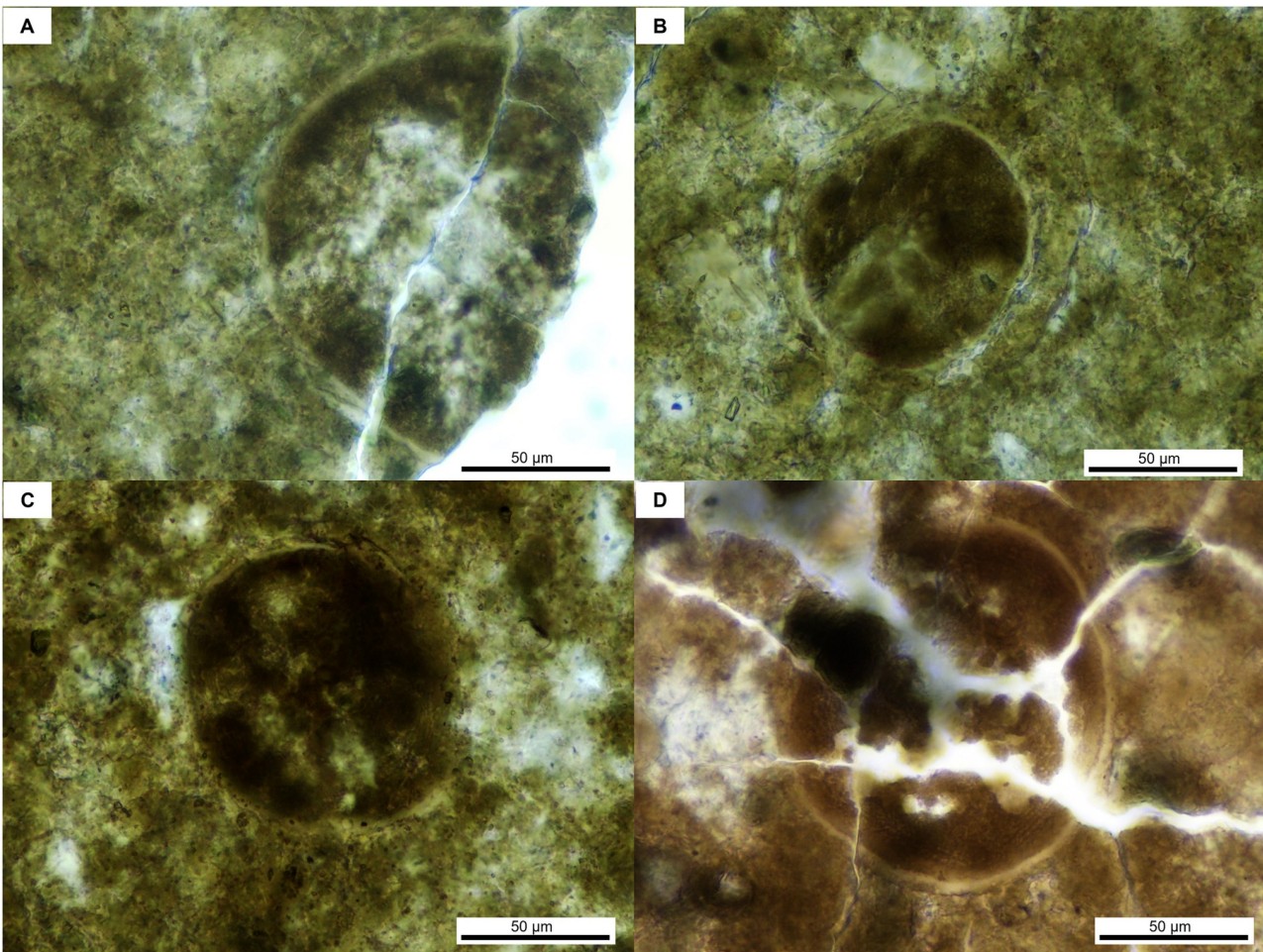

**Fig 6. Parasites of morphotype IV, found in the vertebrate coprolite.**

## Discussion and systematic assessment

The sedimentological conditions of the site where the coprolite was found allowed the preservation of parasites and other organic structures in the coprolite. Microscopic observation of thin sections revealed the highly well-preserved internal structures of the organic structures. This approach had yielded similar results in a study of tapeworm eggs in a shark coprolite [6].

The five observed morphotypes are potentially eggs of parasite. Morphotypes II and IV present some ornamentation on external surfaces. No clearly organized bodies could be defined within these structures. Ornamented surfaces could initially correspond to the sporoderm of fern or moss spores that might have been ingested but not digested by the animal. However, our sections are not showing any fern or moss spore diagnostic features (structures are not triangular or kidney shaped, there is no indication of aperture or sporogenesis scars). The wrinkled surface could correspond to the diagnostic surface of nematode eggs. Using chemical methods may have provided information the external characteristic ornamentation, as in the studies of fossilized Ascaridida eggs using chemical methods [3, 5, 8, 45], but they cannot provide important information on internal structures.

Morphotypes I, III and V apparently do not present trace of ornamentation and are likely not plant spores but rather parasite elimination form through the feces of their hosts. A large

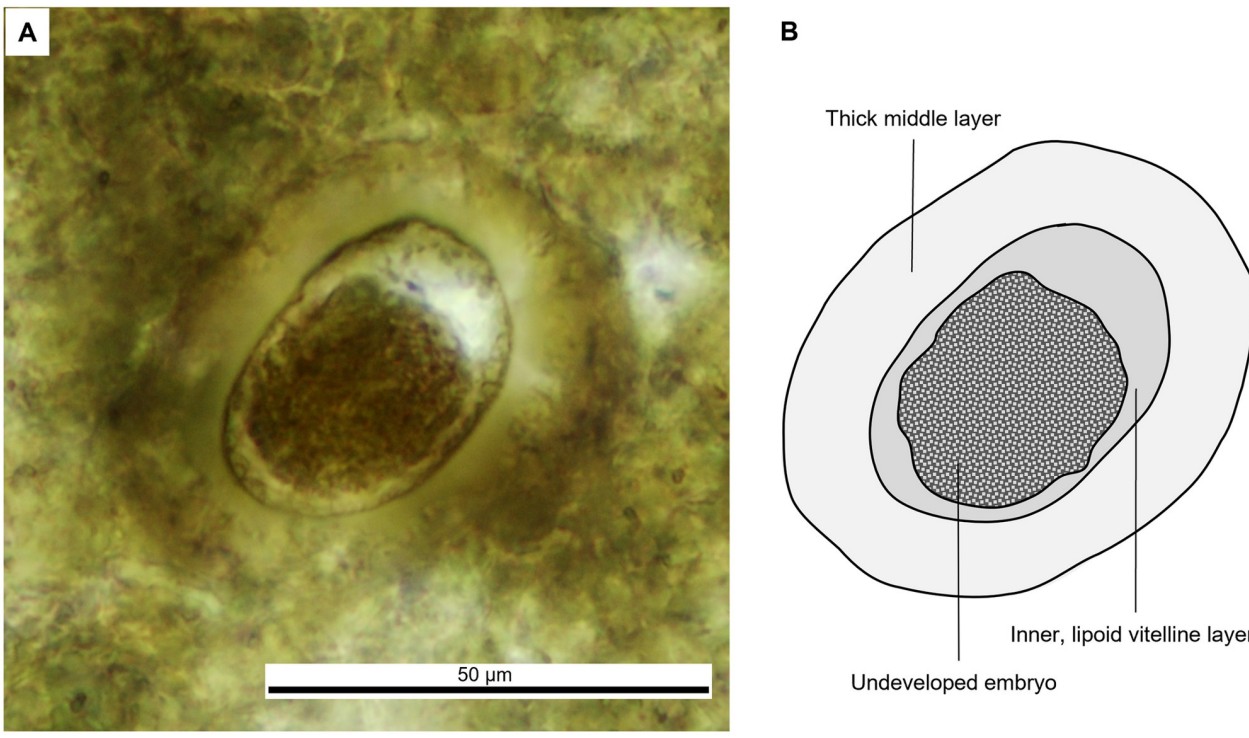

**Fig 7. Parasite of morphotype V, found in the vertebrate coprolite.**

number of parasite groups carry out these elimination steps (protists and helminths). Morphotype I is relatively small and could potentially correspond to unicellular cyst (i.e., Coccidia). Protist cysts have been found in terrestrial coprolites dating back to the early Cretaceous period [45]. Oocysts of coccidians, such as *Eimeria*, overlap in size range with this morphotype [46, 47]. Morphotype III shows a well-developed shell and organized bodies within the shell. It could be a nematode egg with a developed embryo, but confirmation requires other diagnostic characters.

The ellipsoid shape and thick wall of morphotype V are diagnostic traits of eggs of Ascaridida nematodes [48]. Parasites of this order are commonly found in terrestrial vertebrates such as fishes, amphibians, reptiles, birds, mammal-like reptiles, and mammals [3, 5, 8, 49, 50]. Fossil eggs of nematodes have been ascribed to the genus *Ascarites*. The earliest fossil record of Ascarididae eggs, ascribed to *Ascarites rufferi* is Triassic and comes from a cynodont coprolite from Rio Grande do Sul State in Brazil [5]. Other Mesozoic findings are *Ascarites gerus* and *Ascarites priscus* [45] from Early Cretaceous archosaur coprolites (iguanodontian dinosaur) in Belgium, and Ascaridida eggs discovered in Crocodyliformes coprolites from the Early Cretaceous in Brazil [3]. Morphotype V differs from *Ascarites priscus* by its smoother shell and slightly larger size and by its more homogeneous vitelline layers, it differs also from *Ascarites gerus* and *A rufferi* by its smooth and thicker shell. Other individual eggs must be found and studied using both thin sections and chemical techniques to be able to create a new taxon.

The cylindrical and curved shape of the coprolite and the absence of prey remains are typical of the Crurotarsi, especially crocodile-like animals [3, 20, 51, 52]. Crocodiles are not known in Huai Hin Lat but abundant vertebrate assemblages have been discovered in several outcrops of the Huai Hin Lat Formation including actinopterygian fishes [25, 34, 35], lungfish [36, 37],

temnospondyls [25, 27, 38, 39], turtles [40, 41], and phytosaurs [42]. The studied coprolite was therefore likely produced by a crocodile-like reptile, possibly a phytosaur, a reptile that evolved convergently with crocodilians, and whose tooth and bone remains have been found in the same formation [25, 42, 43].

The discovery of at least six parasites with at least five different morphotypes in a single coprolite suggests that multi-parasite infection was common had already diversified by the late Triassic. The presence of the Ascaridida eggs and the evidence for multi-infection found in the coprolite can presumably be explained by the predatory habits of the host, which would have been parasitized by feeding on parasitized fishes, amphibians, or other reptiles [3, 4, 18].

## Conclusion

Parasites of several species, including Ascaridida eggs were found in a coprolite probably produced by a crocodile-like reptile and possibly a phytosaur. This is therefore the first discovery of Ascaridida eggs and evidence of multi-infection in a host assignable to the Crurotarsi from the Late Triassic of Asia.

## Acknowledgments

We would like to thank all staff of the Palaeontological Research and Education Centre (PRC) of Mahasarakham University who took part in the fieldwork.

## Author Contributions

**Conceptualization:** Thanit Nonsrirach, Serge Morand, Alexis Ribas, Julien Claude.

**Data curation:** Komsorn Lauprasert.

**Formal analysis:** Thanit Nonsrirach, Serge Morand, Sita Manitkoon, Julien Claude.

**Funding acquisition:** Sita Manitkoon, Julien Claude.

**Investigation:** Thanit Nonsrirach, Komsorn Lauprasert.

**Methodology:** Thanit Nonsrirach, Alexis Ribas, Komsorn Lauprasert.

**Resources:** Komsorn Lauprasert.

**Supervision:** Serge Morand, Julien Claude.

**Visualization:** Thanit Nonsrirach.

**Writing – original draft:** Thanit Nonsrirach, Sita Manitkoon.

**Writing – review & editing:** Serge Morand, Alexis Ribas, Julien Claude.

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
