## [Decision Letter · Decision Letter 0]

14 Jun 2023

First discovery of parasite eggs in a vertebrate coprolite of the Late Triassic in Thailand

PONE-D-23-16452

Dear Dr. Nonsrirach,

We’re pleased to inform you that your manuscript has been judged scientifically suitable for publication and will be formally accepted for publication once it meets all outstanding technical requirements.

Kind regards,

Jean-lou Justine, DrSc

Academic Editor

PLOS ONE

Additional Editor Comments:

The new version was sent to some of the reviewers of the first version.

I believe that the new version ks now acceptable for publication.

Reviewers' comments:

Reviewer's Responses to Questions

**Comments to the Author**

1. Is the manuscript technically sound, and do the data support the conclusions?

Reviewer #1: Yes

2. Has the statistical analysis been performed appropriately and rigorously? 

Reviewer #1: N/A

3. Have the authors made all data underlying the findings in their manuscript fully available?

Reviewer #1: Yes

4. Is the manuscript presented in an intelligible fashion and written in standard English?

Reviewer #1: Yes

5. Review Comments to the Author

Reviewer #1: I don't have any additional comments to make. The authors' responses are clear, balanced and justified in the light of previous comments and suggestions (especially regarding the debate between coccidia oocyst and nematode egg).

6. PLOS authors have the option to publish the peer review history of their article (what does this mean?). If published, this will include your full peer review and any attached files.

Reviewer #1: No

---

## [Editor Report · Acceptance letter]

11 Jul 2023

PONE-D-23-16452 

First discovery of parasite eggs in a vertebrate coprolite of the Late Triassic in Thailand 

Dear Dr. Nonsrirach:

I'm pleased to inform you that your manuscript has been deemed suitable for publication in PLOS ONE. Congratulations! Your manuscript is now with our production department. 

Kind regards, 

on behalf of

Professor Jean-lou Justine 

Academic Editor

PLOS ONE